# Antinociceptive Effects of Aaptamine, a Sponge Component, on Peripheral Neuropathy in Rats

**DOI:** 10.3390/md21020113

**Published:** 2023-02-04

**Authors:** Chun-Sung Sung, Hao-Jung Cheng, Nan-Fu Chen, Shih-Hsuan Tang, Hsiao-Mei Kuo, Ping-Jyun Sung, Wu-Fu Chen, Zhi-Hong Wen

**Affiliations:** 1Department of Anesthesiology, Division of Pain Management, Taipei Veterans General Hospital, Taipei 112201, Taiwan; 2School of Medicine, National Yang Ming Chiao Tung University, Taipei 112304, Taiwan; 3Department of Marine Biotechnology and Resources, National Sun Yat-Sen University, Kaohsiung 804201, Taiwan; 4Department of Surgery, Division of Neurosurgery, Kaohsiung Armed Forces General Hospital, Kaohsiung 802301, Taiwan; 5Institute of Medical Science and Technology, National Sun Yat-Sen University, Kaohsiung 804201, Taiwan; 6National Museum of Marine Biology and Aquarium, Pingtung 944401, Taiwan; 7Graduate Institute of Natural Products, Kaohsiung Medical University, Kaohsiung 807378, Taiwan; 8Department of Neurosurgery, Kaohsiung Chang Gung Memorial Hospital and Chang Gung University College of Medicine, Kaohsiung 833401, Taiwan

**Keywords:** aaptamine, neuropathic pain, angiogenesis, LDHA

## Abstract

Aaptamine, a natural marine compound isolated from the sea sponge, has various biological activities, including delta-opioid agonist properties. However, the effects of aaptamine in neuropathic pain remain unclear. In the present study, we used a chronic constriction injury (CCI)-induced peripheral neuropathic rat model to explore the analgesic effects of intrathecal aaptamine administration. We also investigated cellular angiogenesis and lactate dehydrogenase A (LDHA) expression in the ipsilateral lumbar spinal cord after aaptamine administration in CCI rats by immunohistofluorescence. The results showed that aaptamine alleviates CCI-induced nociceptive sensitization, allodynia, and hyperalgesia. Moreover, aaptamine significantly downregulated CCI-induced vascular endothelial growth factor (VEGF), cluster of differentiation 31 (CD31), and LDHA expression in the spinal cord. Double immunofluorescent staining showed that the spinal VEGF and LDHA majorly expressed on astrocytes and neurons, respectively, in CCI rats and inhibited by aaptamine. Collectively, our results indicate aaptamine’s potential as an analgesic agent for neuropathic pain. Furthermore, inhibition of astrocyte-derived angiogenesis and neuronal LDHA expression might be beneficial in neuropathy.

## 1. Introduction

Aaptamine (Figure 1) is an alkaloid composed of amines and halogenated cyclic nitrogen. It was first isolated from sponge *Aaptos aaptos* by Nakamura et al. in the South China Sea in 1982 [1]. Aaptamine and its derivatives exert anticancer, antioxidant, and antiviral activity [2,3]. Johnson et al. (2017) found that aaptamine has delta-opioid (DOR) and mu-opioid receptor (MOR) agonist activities, with EC50 of 5.1 μM and 10.1 μM, respectively [4]. A previous study demonstrated aaptamine’s antidepressant activity by decreasing the immobility time of mice in the forced swim test [5]. Moreover, knocking out DOR increases depression-like behavior [6]. Aaptamine presented antidepressant-like activity in wide type but not delta-KO mice [4]. The above results indicate that aaptamine might be a marine natural DOR agonist, and that active DOR have the ability to alleviate neuropathic pain [7].

Nowadays, neuropathic pain affects approximately 6–7% of the general population (over 450 million people in the world), and its incidence is still increasing [8,9]. Neuropathic pain is defined as pain caused by a lesion or disease of the somatosensory system [10]. People with neuropathic pain have allodynia (pain evoked by non-painful stimuli) and hyperalgesia (increased response to painful stimuli), and they might develop nociceptive sensitization and resistance to most analgesics [11]. According to previous research, few current treatments can effectively alleviate the symptoms of neuropathic pain [10].

Glycolysis transforms glucose to pyruvate through multiple cytoplasmic enzymes producing two molecules of ATP. In stress conditions such as inflammation or hypoxia, cells prefer to utilize glycolysis as an energy source rather than the oxidative phosphorylation (OXPHOS) in mitochondria [12,13,14]. Previously, amelioration of glycolysis processes has been shown to improve pain [15,16]. Lactate dehydrogenase A (LDHA), a glycolytic enzyme, converts pyruvate to lactate, which is released to the extracellular space and leading to acidification [17]. Extracellular lactate accumulation influences the development of nociception [18]. In addition, LDHA can translocate into the nucleus not only in response to oxidative stress, but also upregulating vascular endothelial growth factor (VEGF) gene expression [19,20]. Although oxidative stress and VEGF-induced angiogenesis are known to contribute to nociception [21,22], the role of spinal LDHA in nociceptive sensitization remains unclear.

To date, aaptamine and its derivatives have been shown to have anti-HIV, antifungal, antiphotoaging, anti-infective, antifouling, antidepressant, antiviral, antimalarial, and cytotoxic activities [2,3]. However, no previous research has investigated the antinociceptive effects of aaptamine on neuropathic pain. We hypothesized that aaptamine’s analgesic effects in neuropathic pain are likely associated with its inhibitory effects on activated angiogenesis and neuronal LDHA expression in the spinal cord.

## 2. Results

### 2.1. Acute Analgesic Effect of Aaptamine on CCI-Induced Nociceptive Sensitization

The rats were divided into three groups and intrathecally (i.t.) administered with specific doses of 5, 30, and 100 μg of aaptamine, respectively. The nociceptive behavior of thermal hyperalgesia was assessed 30, 60, 90, 120, 150, and 180 min after aaptamine administration through paw withdrawal latency (PWL). The PWL was then converted to the percentage of maximum possible effect (%MPE) (Figure 2B). Our evaluation revealed that the selected doses of 5, 30, and 100 μg presented different durations of analgesic effect of 90, 150, and 150 min, respectively, on CCI-induced nociception. However, the variance in the group administered with 100 μg of aaptamine was the largest, and the rats presented some abnormal motor behavior. The findings provided for a future dose selection of aaptamine to investigate the preventative effects of aaptamine on antinociception and its possible mechanism in CCI rats.

### 2.2. Intrathecal Aaptamine Injection Attenuated CCI-Induced Nociceptive Sensitization

As shown in Figure 3, rats were separated into three groups: control, CCI, and CCI + aaptamine, respectively. Nociceptive behavior, thermal hyperalgesia (paw withdrawal latency, PWL) (Figure 3A), and mechanical allodynia (paw withdrawal threshold, PWT) (Figure 3C) were assessed every two days after CCI surgery. To verify whether intrathecal (i.t.) aaptamine could attenuate CCI-induced nociceptive pain, we injected 30 μg aaptamine per day in the CCI + aaptamine group by i.t. Compared with the CCI group, the CCI + aaptamine group showed significant improvement in PWT and PWL since the day after CCI surgery. In Figure 3B,D, we also demonstrate that aaptamine has an analgesic effect through the maximum possible effect (MPE) of thermal hyperalgesia and mechanical allodynia, respectively. These results indicate that aaptamine is effective in alleviating nociceptive sensitization in CCI rats.

### 2.3. Aaptamine Decreased CCI-Induced Upregulation of Angiogenesis Factors in the Ipsilateral Lumbar Spinal Cord Dorsal Horn

VEGF and CD31 were labeled with anti-VEGF and anti-CD31, respectively (Figure 4A,C). The CCI group expressed significantly more VEGF than the control group. Further, compared with the CCI group, VEGF expression decreased significantly in the CCI + aaptamine group (Figure 4B). The effect of aaptamine on CD31 expression was similar to that of VEGF. The CCI group expressed significantly more CD31 than the control group (Figure 4D). After aaptamine injection, CD31 expression in the CCI + aaptamine group decreased compared to the CCI group. These results demonstrate that aaptamine could decrease the expression of two angiogenesis factors.

### 2.4. Aaptamine Decreased Spinal Astrocytic VEGF Expression after CCI

We then investigated the cellular distribution of VEGF in the ipsilateral lamina I-III of the lumbar spinal cord dorsal horn by immunofluorescence staining of VEGF (red) and three cell-specific markers (green): glial fibrillary acidic protein (GFAP) (for astrocytes), OX42 (for microglia), and NeuN (for neurons), respectively. As shown in Figure 5A, VEGF co-localized with GFAP (yellow) and was expressed at lower levels on microglia (Figure 5B) and neurons (Figure 5C). Pearson’s correlation coefficient analysis was employed to measure quantification (Figure 5D). The Pearson’s coefficient values for VEGF versus GFAP in the control group, the CCI group, and the CCI + aaptamine group are 0.75, 0.82, and 0.70, respectively. The Pearson’s coefficient values for VEGF versus OX42 in the control group, the CCI group, and the CCI + aaptamine group are −0.01, −0.27, and −0.36, respectively. The Pearson’s coefficient values for VEGF versus NeuN in the control group, the CCI group, and the CCI + aaptamine group are −0.04, −0.07, and −0.11, respectively. Pearson’s coefficient values <0.1 were interpreted as no co-localization, and >0.5 were near maximal co-localization. Furthermore, the co-localization of VEGF and GFAP was increased in the CCI group but decreased in the CCI + aaptamine group (Figure 5A,D). As a result, CCI-induced VEGF was mostly expressed in astrocytes in the spinal cord, and aaptamine attenuated the upregulation of spinal astrocytic VEGF after CCI.

### 2.5. Aaptamine Downregulated CCI-Induced Spinal LDHA Expression

We then investigated LDHA expression in the ipsilateral lumbar spinal cord dorsal horn by immunofluorescence staining (Figure 6A). As shown in Figure 6B, LDHA showed low basal expression in the control group. In comparison, LDHA levels increased in the CCI group. However, LDHA expression decreased in the CCI + aaptamine group. Quantitative analysis showed that CCI significantly upregulated LDHA expression, an effect inhibited by i.t. aaptamine administration.

### 2.6. Aaptamine Decreased Spinal Neuronal LDHA Expression after CCI

We investigated the cellular distribution of LDHA in the ipsilateral lamina I-III of the dorsal horn spinal cord by immunofluorescence staining of LDHA (red) and three cell-specific markers (green): GFAP, OX42, and NeuN. As shown in Figure 7A,B, LDHA was expressed at lower levels on GFAP and OX42 but majorly expressed on NeuN (yellow) (Figure 7C). The quantification was measured through Pearson’s correlation coefficient (Figure 7D). The Pearson’s coefficient values for LDHA versus GFAP in the control group, the CCI group, and the CCI + aaptamine group are −0.47, −0.33, and −0.29, respectively. The Pearson’s coefficient values for LDHA versus OX42 in the control group, the CCI group, and the CCI + aaptamine group are −0.07, −0.08, and −0.21, respectively. The Pearson’s coefficient values for VEGF versus NeuN in the control group, the CCI group, and the CCI + aaptamine group are 0.56, 0.71, and 0.55, respectively. The co-localization of LDHA and NeuN was increased in the CCI group but decreased in the CCI + aaptamine group (Figure 7C,D). As a result, CCI-induced LDHA was expressed primarily in neurons of the spinal cord, and aaptamine attenuated the upregulation of spinal neuronal LDHA after CCI.

### 2.7. Aaptamine Attenuated Neuronal LDHA Translocation in CCI

A previous study indicated that LDHA could translocate into the nucleus [19]. To verify whether aaptamine attenuated the CCI-induced neuronal nuclear LDHA by stained NeuN (neuron, green), LDHA (red), DAPI (nucleus, blue), and DRAQ5 (nucleus, blue) under immunofluorescent microscopy (Figure 8A) and laser scanning confocal microscopy (Figure 8C,D) analysis. The co-localization of neuronal LDHA and nucleus is indicated with white arrows in Figure 8A. Nuclear LDHA expression was quantified by Pearson’s correlation coefficient analysis (Figure 8B). According to the results of immunofluorescence staining and Pearson’s correlation coefficient analysis, we demonstrated that LDHA could translocate into the neuronal nucleus. Nuclear LDHA expression increased in the CCI group and decreased in the CCI + aaptamine group. Using confocal microscopy, we observed that LDHA co-localized with NeuN (yellow; Figure 8C) and DRAQ5 (purple; Figure 8D) in the CCI group, indicating that CCI-induced upregulation of neuronal nucleus LDHA.

## 3. Discussion

In this research, we used thermal hyperalgesia and mechanical allodynia to present nociceptive sensitization in CCI-induced neuropathy, and i.t. sponge-derived aaptamine administration significantly alleviated these effects (Figure 2 and Figure 3). Currently, there is no gold-standard therapy for neuropathic pain [23], and investigation and discovery of new potential drugs are warranted. Studies have indicated that systemic adverse events are fewer with intrathecally (i.t.) administration than with systemic therapy because the drug is directly delivered to the site of action, and smaller doses can be used in the former [24]. Thus, i.t. administration is an appropriate experimental method to explore the analgesic effects of compounds on neuropathic pain [25,26,27]. Moreover, we found that aaptamine attenuated the CCI-induced upregulation of the spinal angiogenic markers, VEGF, and CD31 (Figure 4). Similar to our previous study [21], we found that VEGF expresses majorly on astrocytes (Figure 5), and LDHA mainly expresses on neurons in the ipsilateral spinal cord (Figure 7). Additionally, aaptamine could inhibit LDHA upregulation and nuclear translocation in neurons in the neuropathic spinal cord (Figure 6 and Figure 8). It has been known that the gray matter lamina l-lll of the spinal cord dorsal horn primarily involves in nociceptive sensitization [28,29,30]. Several previous papers have demonstrated that spinal immunofluorescence can describe the changes in target proteins in specific central nervous system (CNS) regions [26,31,32,33]. It is very difficult to isolate gray matter of the spinal cord dorsal horn for Western blot or Q-PCR analysis. The immunofluorescence technique can focus on lamina I to III of gray matter in the spinal cord dorsal horn and also determine which cell type is involved in regulating the target protein.

Angiogenesis plays an essential homeostasis role in tissue repair and organ regeneration by forming new blood vessels [34]. Previous studies have found the upregulation of angiogenesis in the neuropathic state [21,35]. VEGF has multiple effects on many cell types and mainly acts on endothelial cells [36]. CD31, known as PECAM-1, mainly expresses on the cell surface in new blood vessels influencing angiogenesis, platelet function, and inflammation in blood vessels [37]. VEGF and CD31 are recognized endothelial cell markers used to identify vascular density and angiogenesis [37,38,39]. VEGF and CD31 upregulation are crucial in the molecular pathogenesis of tumor growth, metastasis, and retinopathy [40]. Moreover, recent studies found a significant upregulation of spinal VEGF protein expression in neuropathological conditions, including neuropathy, experimental autoimmune encephalomyelitis (EAE), and spinal cord injury [35,41,42]. Increasing CD31 expression in the spinal cord has also been observed in spinal cord injury rats and EAE mice [35,43]. Similar to previous research, the present findings showed upregulation of spinal VEGF in neuropathic rats (Figure 3).

Several pro-inflammation cytokines, such as TNF-α, TGF-β, IL-8, and MMP-9, not only promote angiogenesis but also contribute to neuroinflammation and nociceptive sensitization [35,44,45,46]. Croll et al. (2004) found that direct VEGF administration would induce angiogenesis and inflammation in the CNS [47]. As a result, upregulating VEGF expression would lead to angiogenesis, inflammation, and pain. An anti-angiogenesis analog, fumagillin, attenuates angiogenesis and nociception in osteoarthritic animals by inhibiting VEGF expression [48]. Our previous study found that i.t. fumagillin and anti-VEGF antibodies inhibit neuropathy-induced nociceptive sensitization, new spinal vessel formation, and neuroinflammation [21]. Our present findings show that aaptamine attenuates nociception and downregulates the expression of spinal VEGF and CD31 in neuropathic rats (Figure 3). Thus, the analgesic effects of aaptamine include regulating spinal VEGF expression.

In diabetic neuropathic and fibromyalgia patients, cerebral blood flow increases significantly in the right anterior cingulate cortex and anterior/middle cereal arteries, respectively [49,50]. A possible reason for neuropathy-induced spinal angiogenesis is increasing oxygen and nutrient delivery to activate nociceptive sensitization and neuroinflammation. I.t. aaptamine administration might inhibit spinal angiogenesis resulting in attenuation of nociception by decreasing the energy supply from new vessels. Furthermore, VEGF was found to be mainly expressed in CNS astrocytes [51,52,53]. Furthermore, neuropathological insults upregulate astrocytic VEGF expression in the spinal cord [21]. The present results are in agreement with previous literature. CCI-induced astrocytic VEGF upregulation but not on microglia and neuronal cells (Figure 4). We suggest the potential pathway of i.t. aaptamine might inhibit neuropathy-induced spinal angiogenesis by inhibiting astrocytic VEGF release.

Under neuropathological circumstances, the metabolic pathway might change to anaerobic glycolysis in sensitized neurons and activated glial cells for responding to the CNS disorder and fitting the change of energy demand [12,54,55]. Previous studies have indicated that neuropathy causes significant differences in metabolic rate and glucose consumption in CNS [56,57]. LDHA, a glycolytic enzyme, plays a crucial role in fulfilling the energy requirement, regenerating NAD^+^ from NADH and transforming pyruvate into lactate [58,59]. Lactate accumulation is a crucial feature of inflammation [60]. Thus, enhancing LDHA expression, the lactate would accumulate in extracellular space and cause acidification, leading to neurodegeneration and nociception [17,61]. We were the first to find upregulation of spinal LDHA expression in CCI rats. This finding has important implications as spinal LDHA upregulation would increase spinal extracellular lactate concentration, which would contribute to nociceptive transmission, resulting in hyperalgesia and allodynia in neuropathic pain [62]. Therefore, since i.t. aaptamine inhibits spinal LDHA expression (Figure 5), it may decrease lactate production, resulting in nociceptive inhibition.

LDHA exists in two forms, tetramer and dimer are enzymatic active and transcription factors, respectively [19,63,64]. A number of studies have illustrated that the accumulation of reactive oxygen species (ROS) in the spinal cord plays an important role in the development/maintenance of neuropathic pain [65,66,67]. Neuropathic pain is also accompanied by overexpression of HIF1α by ROS, hypoxia, and ischemia stimulation [68,69,70]. Previous studies have demonstrated that HIF1α signaling and ROS accumulation would regulate neuronal LDHA translocation [19,71,72,73]. Moreover, LDHA, as a transcription factor, can regulate VEGF gene expression and promote angiogenesis [74]. In the present study, we found that neuronal LDHA translocated into the nucleus in a neuropathic state (Figure 7). To our knowledge, this is the first study to report neuropathy-induced LDHA upregulation in the nucleus of spinal neurons. Reducing LDHA translocation into the nucleus and regulating its expression could mediate the analgesic effects of aaptamine on neuropathy. We propose that aaptamine produces anti-nociception through mechanisms other than its OR-related activities. The marine sponge-derived natural compound aaptamine has an analgesic effect on neuropathy in rats, possibly by regulating LDHA and angiogenesis.

Three major distinct opioid receptor (OR) types, MOR, DOR, and kappa-opioid receptor (KOR), have been identified. Among these three kinds of OR, MOR activation is most relevant in anti-nociception [75,76]. Morphine is the most common agonist of MOR, which has been found for a long time, and its derivates were the major therapeutic drugs for pain killer in history [77,78,79]. However, the utility of morphine and its derived drugs showed the increasing harmful adverse effects clinically [80]. Cooper et al. also demonstrated that MOR presents low analgesic effects in patients with neuropathic pain [81]. On the other side, DOR shows an analgesic effect on neuropathy [7]. In DOR knock-out (KO) mice, no significant changes are found in pain perception after acute noxious, such as stimuli, thermal, mechanical, or chemical stimulation, but sensitivity to thermal and mechanical stimuli is enhanced in inflammatory and neuropathic pain models [6,82,83,84]. In the inflammation model induced by complete Freund’s adjuvant, the activated spinal DOR is able to attenuate hyperalgesia [85,86,87]. Previous studies illustrated that the potency and analgesic effect of DOR agonist in the chronic pain model is more important than in the acute pain model [88]. Johnson et al. (2017) indicated that aaptamine and its derivates, dimethyl (oxy)-aaptamine and 9-demethyl-aaptamine displayed low micromolar dual DOR and MOR agonist activity [4]. Aaptamine showed antidepressant-like activity in wild type but not DOR KO mice [4,5]. Aaptamine is a full agonist at the DOR and only weak partial agonists at the MOR (Emax < 10% at 50 μM) and KOR activity [4]. Schoos et al. demonstrated that DOR could regulate the expression of HIF1α, the upstream of angiogenesis and LDHA [59,89]. The activation of HIF1α also participates in neuropathy-induced nociception [68]. Thus, we suggest that the potential analgesic pathway of the small molecular marine-derived natural compound, aaptamine, can activate DOR to downregulate the expression of LDHA and angiogenesis in CNS. However, the exact mechanism remains unclear and is worth further investigation.

Over 68% of currently approved small-molecule drugs were originally discovered in natural sources [90]. In developing new drugs derived from natural sources, a critical challenge is finding a sustainable supply of compounds, usually present in low amounts and/or difficult to isolate or synthesize. Aaptamine was first isolated from a marine sponge, *Aaptos aaptos*, in the South China Sea in 1982 [1]. Up to the present, aaptamine is still isolated from the wild type of marine sponge *aaptos* sp., which is hard to culture in a tank. Previous studies indicated that aaptamine and its derivatives have anti-HIV, antiviral, anti-infective, antifungal, antiphotoaging, antifouling, antidepressant, antimalarial, and cytotoxic activities [2,5,91,92,93,94,95]. Previous studies also indicated that the cytotoxic effect of aaptamine (IC_50_ > 88 μM) is lower than its derivatives in three breast cancer cells [96]. Moreover, we preliminary evaluated the cytotoxicity of aaptamine in glioblastoma cell lines (GBM 8401, U87 MG, U138 MG, and T98G) and ovarian cancer cell lines (PA-1 and SKOV-3) through MTT assay. The 24 h cytotoxicity of half-maximal concentration (IC_50_) values of aaptamine on the glioblastoma and ovarian cancer cell lines were approximately 50 μM and >200 μM, respectively. The low toxicity of aaptamine makes itself more suitable for the future development of a drug for non-cancerous diseases such as neuropathological disorders. According to our unpublished observation, the concentrations of both aaptamine and its derivate, isoaaptamine, in the sponge are approximately 2 g/kg in wet weight. Moreover, Gao et al. established the chemical method to synthesize aaptamine and its derivates [97]. However, we cannot evaluate the cost of commercial manufactural synthesis of aaptamine in the chemical method. The aaptamine is worth utilizing cultivation for future drug development. Leal et al. suggested that aquaculture of marine organisms could produce the biomass needed to enable the early stages of drug discovery based on marine natural products [98]. The homogeneous environmental aquaculture system can be used continuously to produce potential bioactive candidates for further drug development. For sustainable use and to obtain more aaptamine, the sponges will transplant and culture in cultivating tanks at the National Museum of Marine Biology and Aquarium in Taiwan, and they will try to resolve sponge culture dilemmas.

## 4. Materials and Methods

### 4.1. Chemical

Freeze-dried and sliced bodies (wet/dry weight = 4.8/2.5 kg) of the sponge specimen Aaptos sp. were extracted with ethanol (EtOH) to provide a crude extract (108.2 g), which was then partitioned between EtOH/n-hexane. The EtOH layer (47.5 g) was applied in a silica gel column chromatography and eluted with gradients of dichloromethane (DCM)/methanol (MeOH) to furnish fractions A~M. To afford aaptamine (130 mg), fraction C (0.5 out of 1.8 g) was chromatographed by normal phase silica gel and eluted with DCM/MeOH (4:1 stepwise to pure MeOH). The structure of aaptamine was elucidated by comparison with the spectroscopic data reported [1].

### 4.2. Animals

Adult male Wistar rats (BioLASCO Taiwan Co., Ltd., Taipei, Taiwan) with body weights ranging between 250 and 285 g were used in this study. Rats were acclimatized for a minimum of six days prior to any procedures and subjected to 12 h light/ dark cycles (lights on at 8:00 a.m.) in temperature-controlled cages at 22 ± 1 °C with a 70% humidity level and free access to food and water. The rats were housed in plexiglass cages and had no restrictions to access food and water. For surgery and drug injection, all rats were anesthetized under 2–3% isoflurane inhalation and aseptic preparation. The postoperative period included the topical application of a povidone-iodine 10% solution and cefazolin (170 mg/kg) intramuscular injection for infection prevention, lidocaine infiltration for pain reduction, and individual adjustments. All experiments and animal use were approved by the Institutional Animal Care and Use Committee of National Sun Yat-sen University (Approval No. IACUC-10447); the use of animals conformed to the Guiding Principles in the Care and Use of Animals, published by the American Physiological Society. In addition, rats were sacrificed after the end of nociceptive behavior experiments, and their spinal cords were examined. Every effort was made to minimize the number and suffering of the animals used. For behavioral analysis, only rats without hematoma or spinal cord injury were tested. Hence, we are confident that the observed biological and biochemical effects were not due to the insertion and treatment of the intrathecal catheter.

### 4.3. Induction of Peripheral Neuropathic Pain and Intrathecal Catheter Implantation

After catheterization, control or CCI surgery was immediately performed on the right sciatic nerve of rats. All surgeries were performed under 2.5% isoflurane anesthesia. As described in Bennett and Xie [99] and our previous studies [32], CCI surgery was performed on the rat’s right sciatic nerve (at mid-thigh level); a 5 mm nerve segment of the sciatic nerve was dissected to place four loosely ligated intestines (4-0 chronic gut). We made a skin incision around the sciatic nerve (at 1 mm intervals) and sutured the skin incision at each layer. In sham-operated rats, surgery was performed to expose only the right sciatic nerve without ligation. We used the i.t. catheter implantation method described in Yaksh and Rudy [100] and our previous studies [101]. Briefly, an i.t. catheter (PE5 tubes: 9 cm, 0.008-inch inner diameter, 0.014-inch outer diameter) was inserted through the atlanto-occipital membrane into the i.t. space (lumbar enlargement) of the spinal cord; externalized and fixed to the cranial aspect of the head. After surgery, rats were replaced in their home cages for a 5-day recovery period. Rats presenting with signs of severe nerve damage or hematoma in cerebrospinal fluid were excluded from the study. In the acute analgesic effect of aaptamine experiment, rats were randomly divided into three groups with six rats in each group as follows: (i) CCI + aaptamine 5 μg; (ii) CCI + aaptamine 30 μg; (iii) CCI + aaptamine 100 μg—the rats received intrathecal (i.t.) aaptamine with given doses. In the preventative analgesic effects of the aaptamine experiment, rats were randomly divided into three groups with six rats in each group, as follows: (i) control group; (ii) CCI group; (iii) CCI + aaptamine group—the rats received intrathecal (i.t.) aaptamine (30 µg/day) for 13 days after CCI. Aaptamine was delivered in 10 µL artificial cerebrospinal fluid, consisting of 2.6 mM K^+^, 21.0 mM HCO_3_^−^, 151.1 mM Na^+^, 1.3 mM Ca^2+^, 3.5 mM dextrose, 0.9 mM Mg^2+^, 2.5 mM HPO_4_^2−^, and 122.7 mM Cl^−^.

### 4.4. Thermal Hyperalgesia

Rats were placed into a compartment of a transparent plastic cage upon an elevated glass platform. Then, we measured thermal hyperalgesia with an IITC analgesiometer (IITC Inc., Woodland Hills, CA, USA) as previously described by Hargreaves et al. [102] and our previous studies [21,103]. The rats’ middle plantar surface of the right hind paw was exposed to a radiant heat source with low-intensity heat (active intensity = 25) through the glass platform. The mean paw withdrawal latency (PWL; in seconds) was averaged from the latency of three positive tests. Licking or rapid paw withdrawal were considered positive features of pain behavior. The PWL was set to a cut-off time of 30 s. PWL was transformed to a percentage of the maximum possible effect (% MPE) using the following formula: % MPE = (post-drug latency–baseline)/(cutoff–baseline) × 100%.

### 4.5. Mechanical Allodynia

Rats were placed into a compartment of a transparent plastic cage upon an elevated metal mesh floor for easily approaching the rats’ hind paws. Then, we measured mechanical allodynia with von Frey filaments (Stoelting, Wood Dale, IL, USA), as previously described by Chaplan et al. [104] and our previous studies [21,103]. The middle of the right hind paw was perpendicularly subjected to a series of von Frey filaments with logarithmically incrementing stiffness (0.2–10 g) until the filaments slightly bowed. The mean paw withdrawal threshold (PWT; in grams) determined using Chaplan’s “up-down” method was averaged from the threshold of three positive tests. Rapid paw withdrawal or licking were considered positive features of pain behavior. The PWT was set to a cut-off weight of 10 g. PWT was transformed to a percentage of the maximum possible effect (% MPE) using the following formula: % MPE = (post-drug threshold–baseline)/(cutoff–baseline) × 100%.

### 4.6. Immunohistofluorescence Assay

The immunohistochemistry protocol and image quantification were performed as in our previous studies [21,32,105]. Briefly, after 14 days post-CCI injury, rats were terminally anesthetized with isoflurane and sacrificed by transcardial perfusion with cold phosphate-buffered saline (PBS) (pH 4.7) containing heparin (200 U/mL) and followed by 4% paraformaldehyde in PBS. Lumbar regions (L4–L6) of the spinal cord were harvested, mounted on a tissue block with OCT (Sakura Finetek, Torrance, CA, USA), cut into 20 µm-thick sections with cryostat Microm HM550 (Waldorf, Germany), mounted serially on microscope slides, and processed for immunofluorescence studies. These sections were incubated with a blocking buffer (10% BSA, 0.3% TritoX-100, 0.05% Tween 20, 1X PBS) for 1 h and incubated overnight at 4 °C with each primary antibody: mouse monoclonal anti-neuronal nuclei Alexa Fluor 488 (NeuN, 1:1000, cat. MAB377X; EMD Millipore, Temecula, CA, USA), mouse monoclonal anti-glial fibrillary acidic protein antibody (GFAP; astrocytic marker, 1:1000 dilution, cat. MAB3402; EMD Millipore, Temecula, CA, USA), mouse monoclonal anti-CD11b antibody (clone OX-42, 1:600 dilution, cat. CBL1512; EMD Millipore, Temecula, CA, USA), mouse monoclonal anti-VEGF antibody (1:200, cat. 05-443; EMD Millipore, Temecula, CA, USA), goat polyclonal anti-CD31 antibody (1:200 dilution, cat. AF3628; R&D Systems, Inc., Minneapolis, MN, USA) and rabbit monoclonal anti-lactate dehydrogenase A (LDHA, 1:800 dilution, cat. MBP2-67483; Novus Biologicals, LLC, Centennial, CO, USA). After incubating in blocking buffer, the sections were incubated with a mixture of Alexa Fluor 488-conjugated anti-mouse IgG antibody (1:1000 dilution), Alexa Fluor 488-conjugated anti-rabbit IgG antibody (1:1000 dilution), Cy^TM^ 3-conjugated anti-mouse IgG antibody (1:800 dilution), Cy^TM^ 3-conjugated anti-rabbit IgG antibody (1:400 dilution), DAPI (1:200 dilution, Invitrogen D21490.), and DRAQ5 (1:200 dilution, Abcam ab108410) for 40 min at room temperature.

The sections were examined by a Leica DM-6000B fluorescence microscope (Leica, Wetzlar, Germany) fitted with SPOT Xplorer Digital integrating camera (Diagnostic Instruments Inc., Sterling Heights, MI, USA) and analyzed by SPOT software 4.6 (Diagnostic Instruments Inc., Sterling Heights, MI, USA). The exposure time was the same for all spinal cord sections on the same slide. Image J software 2.9.0 (National Institutes of Health, Bethesda, MD, USA) was utilized for pixel measurement and analysis. To quantify the immunofluorescent images from lamina I to lamina III in the spinal cord dorsal horn, we calculated the mean values for three rats per group. The image size and image acquisition parameters were maintained constant for all conditions on each side of the spinal cord dorsal horn. Confocal images were captured by Leica TCS SP5II equipped with Leica HyD (Hybrid Detector). Representative images of spinal cord immunostaining were taken at 100× and 400× magnification; co-localization images were taken at 400× or 630× magnification.

### 4.7. Co-Localization Analysis

Co-localization analysis was calculated by Pearson’s correlation coefficient [106], and the quantification was performed by an intensity correlation coefficient-based method applying FIJI, an ImageJ program with a plugin [107]. The background was subtracted from the red and green channels (8 bits each) using Pearson’s correlation plugin [108]. Pearson’s correlation coefficient measures the overlapping ratio between two channels and is not sensitive to differences in signal intensities caused by variations in fluorophores, photobleaching, or different amplifier gain settings in an image [106]. Pearson’s correlation coefficient, also explained by its liner regression with Pearson’s coefficient, values range from 1, indicating the fluorescence intensities of two images are a completely positive correlation, to −1, representing the fluorescence intensities of two images are completely negative correlation, compared to one another [109]. Pearson’s coefficient values <0.1 were interpreted as no co-localization, and >0.5 were near maximal co-localization [110].

### 4.8. Statistical Analysis

Data are presented as means ± standard error of the mean (SEM). Changes in protein levels and immunofluorescence reactivity were shown relative to control levels. For statistical analyses, we calculated the variation between groups by a one-way analysis of variance (ANOVA), examined by a post hoc Tukey test. We defined the statistical significance as *p* < 0.05. Statistical analyses were performed using SigmaPlot Version 12.0 (Systat Software, Inc., San Jose, CA, USA).

## 5. Conclusions

In conclusion, the present studies demonstrated that neuropathy upregulates angiogenesis and neuronal LDHA expression in the spinal cord. The marine sponge-derived natural compound aaptamine has an analgesic effect on neuropathy rats, possibly by regulating LDHA and angiogenesis. The present might help future researchers to understand the mechanism(s), pathway, and effects of aaptamine on neurological disorders.

## Figures and Tables

**Figure 1 marinedrugs-21-00113-f001:**
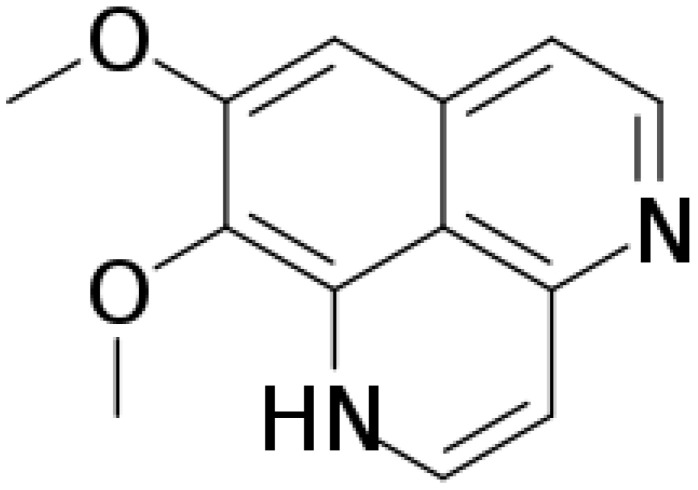
The chemical structure of aaptamine.

**Figure 2 marinedrugs-21-00113-f002:**
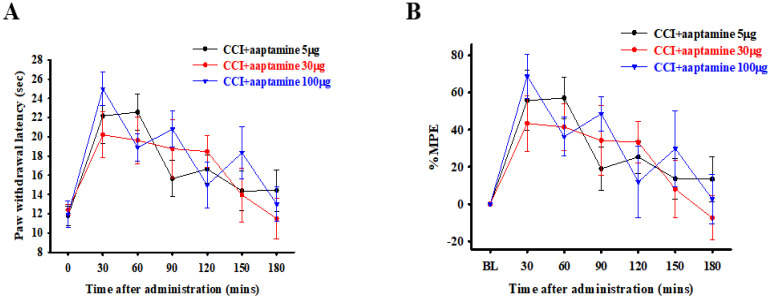
Acute antinociceptive effects of aaptamine in CCI-induced neuropathic rats. Aaptamine was intrathecally administered in three groups of rats in doses of 5, 30, and 100 μg, respectively. Thermal hyperalgesia, induced by CCI, was evaluated 30, 60, 90, 120, 150, and 180 min after aaptamine administration through PWL measured in seconds (**A**) and then converted to %MPE (**B**). (CCI, chronic constriction injury; PWL, paw withdrawal latency; %MPE, maximum possible effect).

**Figure 3 marinedrugs-21-00113-f003:**
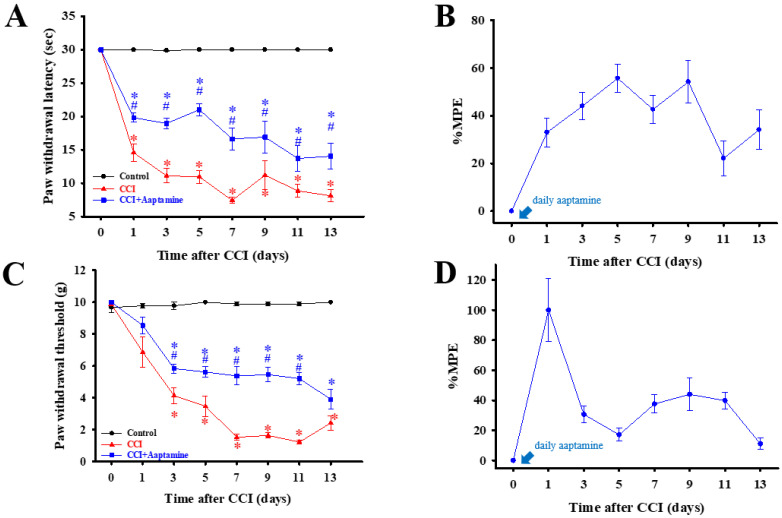
The preventative analgesic effects of aaptamine on CCI-induced neuropathic rats. Intrathecally administered aaptamine (30 μg/day) after CCI. Thermal hyperalgesia and mechanical allodynia, induced by CCI, were evaluated by the paw withdrawal latency (PWL, s) (**A**) and paw withdrawal threshold (PWT, g) (**C**), respectively. PWL and PWT were evaluated into maximum possible effect (MPE), (**B**,**D**), respectively. Data are presented as mean ± standard error (SEM) of PWL and PWT. * and # represent *p* < 0.05 compared with the control and CCI group, respectively. The post hoc Tukey test was used to examine the one-way analysis of variance (ANOVA).

**Figure 4 marinedrugs-21-00113-f004:**
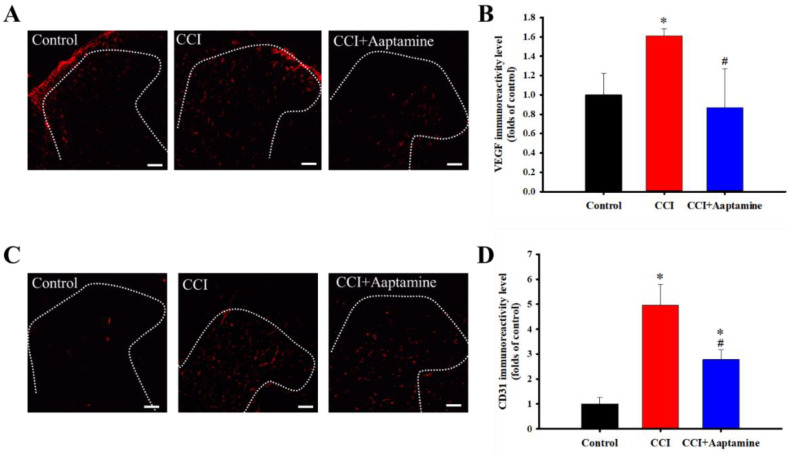
The inhibitory effects of aaptamine on angiogenesis factors in the dorsal lumbar spinal cord after CCI. Lumbar spinal cords were harvested at postoperative day 14 of CCI after receiving the last intrathecal aaptamine injection in the control, the CCI, and the CCI + aaptamine groups. Immunofluorescence images show the vascular endothelial growth factor (VEGF) (red; **A**) and the cluster of differentiation 31 (CD31) (red; **C**) in the ipsilateral spinal cord dorsal horn. The quantification of VEGF (**B**) and CD31 (**D**) immunoreactivity is shown as means ± standard error of the mean (SEM). Aaptamine administration (30 μg/day) attenuated CCI-induced VEGF and CD31 upregulation on the ipsilateral spinal cord dorsal horn. The dotted line represented the gray/white matter boundary. * and # represent *p* < 0.05 compared with the control and the CCI group, respectively. The post hoc Tukey test was used to examine the one-way analysis of variance (ANOVA). Scale bars: 100 μm for all images.

**Figure 5 marinedrugs-21-00113-f005:**
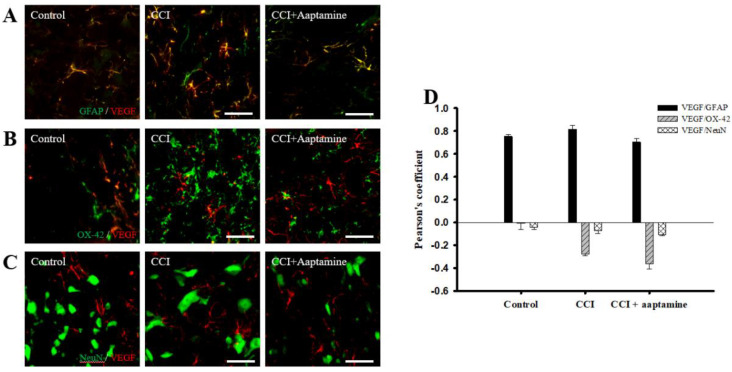
Aaptamine attenuates CCI-induced VEGF upregulation in spinal astrocytes. Lumbar spinal cords were harvested at postoperative day 14 of CCI after the last intrathecal aaptamine injection in the control, the CCI, and the CCI + aaptamine groups. Co-localization of VEGF with cell-specific markers for astrocytes, neuronal cells, and microglia, including GFAP, neuronal nuclear protein (NeuN), and OX-42, respectively. Merged images of double-immunofluorescence staining for VEGF (red) with (**A**) GFAP (green), (**B**) OX42 (green), and (**C**) NeuN (green) in the ipsilateral lumbar spinal cord dorsal horn in the control, CCI, and CCI plus aaptamine groups. VEGF expression was primarily restricted to astrocytes and attenuated by i.t. aaptamine. (**D**) Graphic representation (scatter plot) of Pearson’s correlation coefficient for quantifying the co-localization between anti-VEGF and GFAP, NeuN, or OX-42. Scale bars: 25 μm for all images.

**Figure 6 marinedrugs-21-00113-f006:**
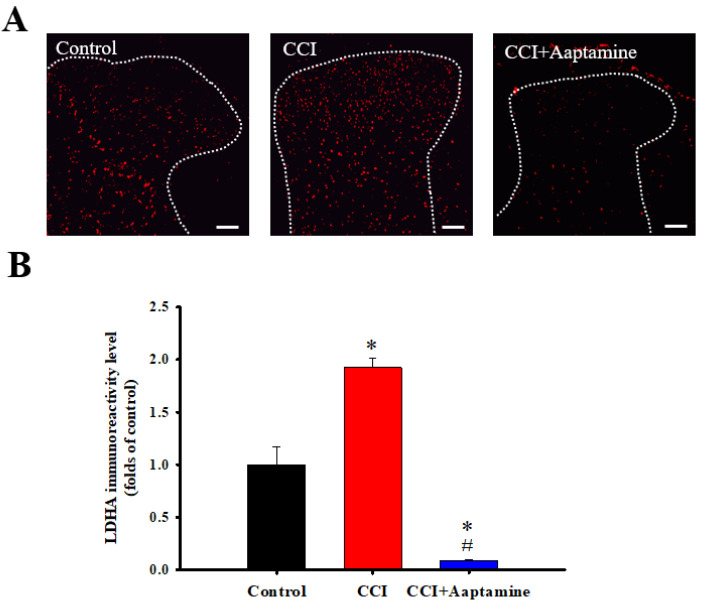
The inhibitory effects of aaptamine on CCI-induced spinal lactate dehydrogenase A (LDHA) upregulation in the dorsal lumbar spinal cord. (**A**) Lumbar spinal cords were harvested at postoperative day 14 of CCI after receiving the last intrathecal aaptamine injection in the control, the CCI, and CCI + aaptamine groups. Immunofluorescence images showing cells labeled with LDHA (red) in the ipsilateral spinal cord dorsal horn. (**B**) The quantification of LDHA immunoreactivity is shown as means ± standard error of the mean (SEM). Spinal LDHA expression was upregulated after CCI and attenuated by i.t. aaptamine administration. The dotted line represented the gray/white matter boundary. * and # represent *p* < 0.05 compared with the control and the CCI group, respectively. A post hoc Tukey test was used to examine the one-way analysis of variance (ANOVA). Scale bars: 100 μm for all images.

**Figure 7 marinedrugs-21-00113-f007:**
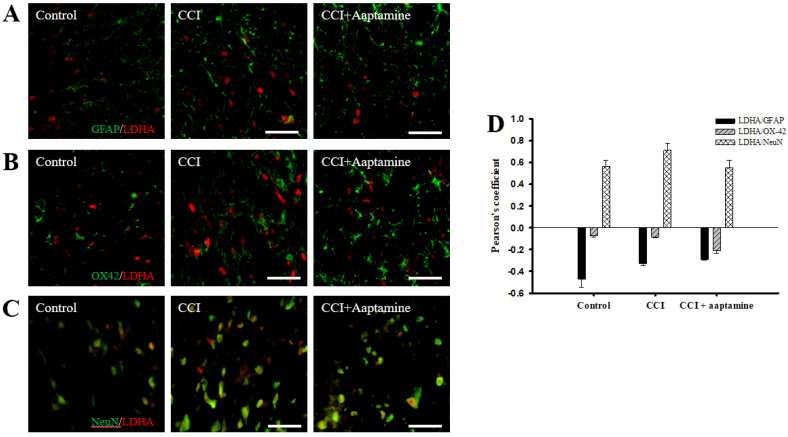
Aaptamine treatment decreases CCI-induced upregulation of neuronal LDHA in the dorsal lumbar spinal cord. Lumbar spinal cords were harvested at postoperative day 14 of CCI after the last intrathecal aaptamine injection in the control, the CCI, and the CCI + aaptamine groups. LDHA co-localization with cell-specific markers for astrocytes, neuronal cells, and microglia are glial fibrillary acidic protein (GFAP), neuronal nuclear protein (NeuN), and OX-42, respectively. Merged images of double-immunofluorescence staining for LDHA (red) with (**A**) GFAP (green), (**B**) OX-42 (green), and (**C**) NeuN (green) in the ipsilateral lumbar spinal cord dorsal horn of the control, CCI, and CCI + aaptamine groups. LDHA expression was primarily restricted to neurons and attenuated by i.t. aaptamine administration. (**D**) Graphic representation (scatter plot) of Pearson’s correlation coefficient for quantifying the co-localization between anti-LDHA and GFAP, OX-42, or NeuN. Scale bars: 25 μm for all images.

**Figure 8 marinedrugs-21-00113-f008:**
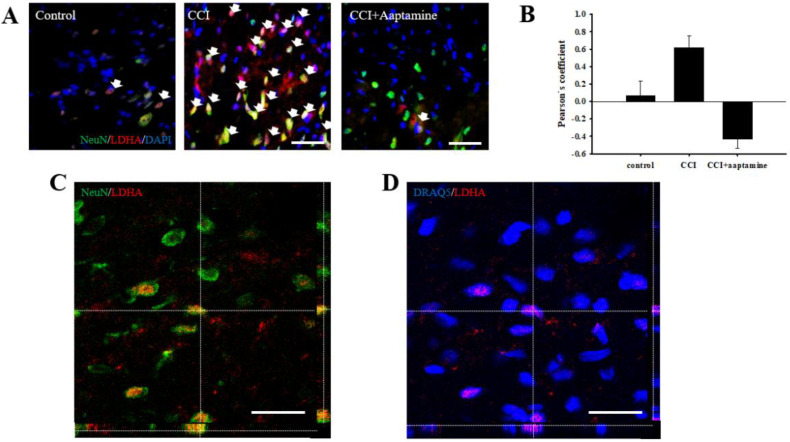
Aaptamine attenuates CCI-induced upregulation of spinal neuronal nuclear LDHA. (**A**) Merged images of triple-immunofluorescence staining for LDHA (red) with NeuN (green) and DAPI (blue) in the lumbar spinal cord dorsal horn of the control group, the CCI group, and the CCI + aaptamine group. The white arrow symbol denotes the LDHA+NeuN+DAPI co-localization. (**B**) The quantification of neuronal nuclear LDHA through Pearson’s correlation coefficient analysis. (**C**) The CCI-induced neuronal LDHA expression image was captured by laser scanning confocal microscopy. The immunofluorescence staining is LDHA (red) and NeuN (green). The images of the ipsilateral lumbar spinal cord reveal the immunoreactivity signal of LDHA to be co-localized with neurons. (**D**) The CCI-induced nuclear LDHA expression image is captured by laser scanning confocal microscopy. The immunofluorescence staining is LDHA (red) and DRAQ5 (blue). The uncropped images of (**C**,**D**) are presented in Appendix A. The images of the ipsilateral lumbar spinal cord reveal the immunoreactivity signal of LDHA to be co-localized with the nucleus. Scale bars: 25 μm.

## Data Availability

Not applicable.

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
