# Peer review of "Antinociceptive Effects of Aaptamine, a Sponge Component, on Peripheral Neuropathy in Rats"

_marinedrugs, 2023, doi:10.3390/md21020113_

Round 1

Reviewer 1 Report

In the manuscript entitled “Antinociceptive effects of aaptamine, a sponge component, on peripheral neuropathy in rats” the authors studied the several effects of the aaptamine, a marine natural compound on chronic constriction injury (CCI)-induced peripheral neuropathic rat model. They investigated the analgetic effects of aaptamine, as well as expression and intracellular localization of angiogenesis factors VEGF and CD31, or LDHA in ipsilateral lumbal spinal cord after aaptamine administration in CCI rats. Also, in Results and Discussion, the authors suggest several molecular mechanisms of aaptamine’s role in antinociception. In my opinion, the study presented in this paper is relevant for neuropathy and for the use of natural sponge compounds as a treatment of neurological disorders. The amount of performed work and the quality of the experiments is suitable for the publication in Marine Drugs.

In order to improve the quality of the manuscript, a major revision would be necessary.

1. Throughout Results the authors comment proteins expression levels based on the result observed by immunofluorescence. However, for detection of protein levels in chosen tissues other methods are more suitable, like Western blot. Authors are encouraged to add these experiments or address in more details why they find immunofluorescent data sufficient for their conclusions regarding protein expression.

2. Section 2.7. Lines 204-208, the authors state: “As is shown in Fig. 7A, nuclear LDHA was upregulated in the CCI group but decreased in the CCI + aaptamine group. With confocal microscopy, we found that LDHA co-localized with NeuN (yellow; Fig. 7B) and DRAQ5 (purple; Fig. 7C) in the CCI group.” However, these sentences do not describe the shown results in Figure 7. in enough details. In addition, important question should be resolved, is the observed effect of downregulation of nuclear LDHA in CCI+aaptamine due to general downregulation of LDHA in the region analyzed, or specifically due to difference in LDHA localization upon aaptamine treatment. Therefore, the appropriate quantification of immunofluoresce should be added (ratio of nuclear LDHA vs. nonnuclear LDHA or similar). Additionally, Western blots of nuclear fractions can be added.

3. The authors should describe obtained quantitative data more scientifically throughout the whole Results (maybe by using p-values), and avoid phrases as “obviously decreased” as in lines 138 and 182.

4. Figure title should include a brief and most important conclusion of the shown figure, and not only the description of the method. Therefore, conclusions written at the end of the figure legend should be moved and combined as a figure title. For example, in Figures 1, 2, 3, 5, 7.

5. Figure 1. The lines connecting data points (CCI+ aaptamine 30 µg vs. CCI+ aaptamine 100 µg) should be more distinctive (maybe in different color).

6. The manuscript needs revision for language and grammar. Some sentences are hard to understand.

Reviewer 2 Report

Firstly, there is no such an abbreviation as delta-OR for delta opioid receptors, similarly form mu opioid. Instead, please correct them into DOR and MOR, respectively.

The Authors writtent that "The compound aaptamine were isolated from Aaptos sp. and their chemical structures". My question is whether this is one compound or at least two? In line with this, please provide some information about the extraction procedure, as this was not included in the main text (methodology).

In the subsection "animals", please state the total number of animals used. In the methodology the Authors stated that the compound was administered at a dose of 30 μg/day, while in the results it was provided that the compound was given at three different doses of 5, 30, and 100 μg. Please verify this. Also, please explain on what basis did the Authors choose to use such doses? It would be nice, if the Authors could provide with EC50.

Since the Authors state that the drug was administered at a specific dose per day, my question is for how many days did the Authors adminstere the drug? Was it acute or chronic adminstration? this should be stated in the methodology.

MPE values and much deeper results description should be given.

Since the Authors administered the compound within 13 days, I wonder whether there is any information regarding  enzymatic stability of the compound. This would improve the paper

The Authors mentioned in the introduction section that the compound has the affinity towards opioidergic receptors. I was wonder whether the Authors have any results demonstrating if such receptors are also involved in the compound-induced effect in neuropathic pain? (any blockade with opioid antagonists?)

Reviewer 3 Report

Antinociceptive effects of aaptamine, a sponge component, on peripheral neuropathy in rats

Sung et al. reported the antinociceptive effect of aaptamine, a well-known natural compound having several pharmacological activities including antimicrobial, anti-depressant, anticancer and so on. In this study the authors suggested that aaptamine showed antinociceptive effects by regulating LDHA and angiogenesis. Background of the study is clear. Methodology has been described appropriately. Results are presented logically and discussion are very clear.  Thus, I would recommend to accept the paper with the following observations:

1.      Did the authors isolate the compound, aaptamine? It would be better if you can show the structure of aaptamine, anywhere in the manuscript.

2.      There are so many abbreviated words in the manuscript that have not been elaborated. For example, although it is well known that OXPHOS mean ‘Oxidative phosphorylation’ , however, it should be defined anywhere in the manuscript. I would like to make a list of abbreviations at the end of the manuscript (before references).

3.      In biology, it is always recommended to use a positive control (standard) in all experiment to ensure that the experiment is functioning. This is the only weak point of this manuscript. The authors did not use any positive control. Explanation on this point is needed.

4.      There are several studies that aaptamine has cytotoxic effect. In order to become a compound to be drug, it is wise to see the effect of the compound on normal cell line. I would suggest the authors to test the compound on normal human cell line to ensure the safety of the compound.

Round 2

Reviewer 1 Report

The authors answered all of my concerns. I consider that the revised manuscript is ready to be accepted.

Kind regards,

Helena Ćetković

Reviewer 2 Report

The Authors have provided additional comments and improved the manuscript. Therefore, in my opinion, it is suitable to be published.